environmental engineering/nanotechnology

biomass carbon, supercapacitor, energy storage devices, polysaccharide, fungal substrate

**Authors for correspondence:**
Mingtang Li
e-mail: limtdoc2008@163.com
Zhiqiang Cheng
e-mail: czq5974@163.com

This article has been edited by the Royal Society of Chemistry, including the commissioning, peer review process and editorial aspects up to the point of acceptance.

# Novel biomass-derived smoke-like carbon as a supercapacitor electrode material

Mingxu Chu[1], Mingtang Li[1], Zhaolian Han[1], Jinshan Cao[1], Rui Li[1] and Zhiqiang Cheng[1,2]

[1]College of Resource and Environment, Jilin Agriculture University, Changchun, Jilin, People's Republic of China
[2]College of Physics, Tsinghua University, Beijing, People's Republic of China

MC, 0000-0003-1457-2467; ML, 0000-0002-5222-866X; ZC, 0000-0003-1156-0540

In this present work, smoke-like carbon was successfully fabricated from a bio-waste fungal substrate crude polysaccharide for the first time. The as-prepared products possess smoke-like structures, ultra-high specific surface area ($S_{BET}$: 2160 m$^2$ g$^{-1}$) and a high content of micropores (microporous surface area of 60%, with a nanopore size of 0.70 nm), which can increase the specific capacitance, representing a wonderful structure for electrochemical energy storage devices. The as-prepared sample displayed an excellent specific capacitance of 152 F g$^{-1}$ at 5 A g$^{-1}$ in the three-electrode configuration and exhibited maximal densities of 6.8–10.2 W h kg$^{-1}$ under power outputs of 253.4–24.3 kW kg$^{-1}$. We believe that this work demonstrates a simple, green and low-cost route by using agricultural residue to prepare applicable carbon materials for use in energy storage devices.

## 1. Introduction

Depletion of fossil fuels and the environmental problems caused by pollution make it imperative to change the current model of energy production and consumption. Electric double-layer capacitors (EDLCs)/supercapacitors [1] are a new type of energy storage device with electrochemical performance between traditional capacitors and batteries that have attracted considerable attention [2]; they have been emerging as an ideal energy storage device for portable electronics, electric vehicles and other high-power applications due to their unique characteristics such as safety, long lifespan, high power density and rapid charge–discharge capability [3,4]. However, they still suffer from low energy

densities as compared to commercial lithium-ion batteries [5], which has significantly limited their further application as primary power sources. The previous studies have shown that the choice and optimization of electrode materials have a great impact on the practical application of supercapacitors.

As is known, carbon materials [6] are the most commonly used electrode materials because of their chemical stability, open porosity and environmental friendliness [7], including activated carbons (ACs), graphene, carbon nanotubes, etc. Among them, the ACs derived from biomass, such as coconut shell [8], corn [9], straw [10], husk [11] or tannins [12], have been extensively applied because of their low cost and low toxicity. Many pre-existing studies have demonstrated that surface microstructures' specific surface area is essential for realizing high electrochemical performance supercapacitors [13]; however, most of the biomass-derived activated carbon prepared by activation post-treatments (physical or chemical activation) may present a limited surface area [14].

Crude polysaccharides [15] have been well studied in the biomedical field [16], which we first tried to apply to the field of biochar materials [17]. Through actual observation, we found that there are still a lot of fungal polysaccharide in the waste fungal substrate (7 wt%); therefore, from these wastes, a large amount of crude fungal polysaccharide can still be extracted using traditional methods [18]. In this work, we obtained a promising electrode material using a fungal substrate crude polysaccharide. The as-prepared products show a large specific surface area ($2160 \, \mathrm{m^2 \, g^{-1}}$) and excellent porosity, with a representative pore size. The material shows outstanding specific capacitance. This approach has great potential for realizing large-scale green and low-cost production of biomass carbon materials for future energy storage applications [19].

In this work, we obtained a promising electrode material using crude polysaccharide extracted from the waste fungal substrate. The as-prepared products showed a large specific surface area ($2160 \, \mathrm{m^2 \, g^{-1}}$) and excellent porosity, with a representative pore size of 0.7 nm. The material shows outstanding specific capacitance in the three-electrode system. This approach is of great potential for realizing large-scale green and low-cost production of biomass carbon materials for future energy storage applications.

# 2. Experiments

## 2.1. Materials

The waste fungal substrates were collected from the Institute of Edible Fungi of Jilin Agricultural University. Nitrogen (greater than 99.999%) was from Changchun Juyang Gas Co., Ltd. Hydrochloric acid (HCl, analytical grade), ethanol (analytical grade), acetone (analytical grade) and potassium hydroxide (KOH, analytical grade) were from Tianjin Chemical Reagent Co., Ltd. N-Methylformamide (NMF) and vinylidene fluoride (PVDF) were from Aladdin. Platinum electrode, mercury oxide electrode and acetylene black (ECP600JD) were from Tianjin Aidahengsheng Technology Development Co., Ltd. Deionized water was used in all experiments.

## 2.2. Extraction of polysaccharide

The extraction of polysaccharide from waste fungal substrates was performed using a method modified from that of Samavati [20] with some improvements. The fungal substrates were ground into a powder form, washed three times with water and air-dried at room temperature for 48 h. The powdered samples were extracted by microwave with a reflux device of 400 W for 30 min.

## 2.3. Preparation of crude polysaccharide carbons

Carbon materials were prepared from crude polysaccharide by the three procedures detailed below.

The crude polysaccharide obtained in step 2.1 was pre-oxidized in a tube furnace (100°C) for 30 min in an air atmosphere with a heat rate of 5°C.

After being pre-oxidized, the carbon precursor was directly calcined at 900°C for 1 h with a heat rate of 5°C min$^{-1}$ under $N_2$ atmosphere. The resultant carbon was denoted as CPC900 (crude polysaccharide carbon 900); for comparison, the carbon precursor was also calcined at different temperatures of 400 and 600°C, denoted as CPC400 and CPC600.

The resultant carbon materials were washed with HCl (5%) and acetone (5%) to remove various inorganic impurities.

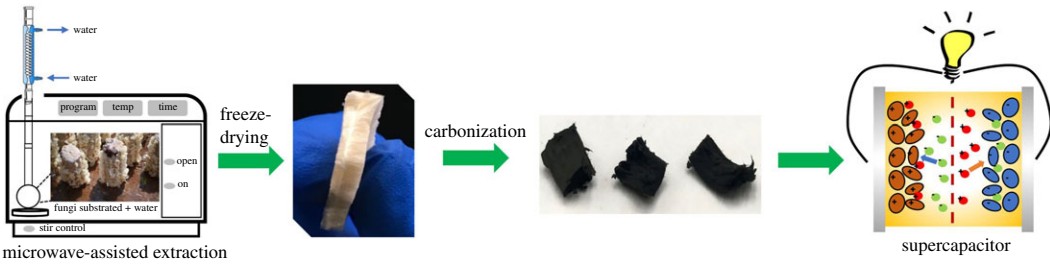

**Scheme 1.** Illustration of the synthesis of the CPCs.

The synthesis is shown in scheme 1. The method of fabrication of electrodes and solid-state symmetric supercapacitors was shown in electrochemical impedance spectroscopy (EIS).

## 2.4. Characterization techniques

Thermogravimetric analysis was carried out in HCT-4 (Hengjiu, Beijing); the crude polysaccharide (8 mg) was heated under $N_2$ atmosphere to the target temperature (1050°C) with a heat rate of 5°C min$^{-1}$. Scanning electron microscopy images (SEM) were operated from SHIMAZU X-550 after metallization. Transmission electron microscope images (TEM) were obtained with TECNAI G2. Raman spectra were operated from a Horiba LabRAM Raman spectrometer. The Raman scattered light was dispersed by a holographic grating, 1200 lines mm$^{-1}$ and was detected by a CCD camera, the wavelength of the laser was 532 nm and filtered at 1% of its nominal power. In order to avoid any damage or heating to the sample, the incident power was controlled at 1 mV. X-ray diffraction spectra were obtained from Bruker AXS D8 Advance, with a scan rate of 2° min$^{-1}$.

Each spectrum was obtained from 0 to 80 target. The nitrogen adsorption isotherms were operated from BeiShiDe 3H-2000PS1. The samples were degassed at 120°C for 24 h prior to any measurement.

## 2.5. Electrochemical measurements

Electrochemical characterizations were performed using a CHI760E workstation. Cyclic voltammetry (CV) and galvanostatic charge–discharge tests (GCD) were carried out in a three-electrode system. Six molar KOH was used as an aqueous electrolyte, with platinum used as the counter electrode, Hg/HgO was used as the reference electrode and the as-prepared products on nickel foam with active materials (3.0 mg) were used as the working electrode. EIS spectra were carried out in a symmetric two-electrode system with the same electrolyte. CV tests were carried out in the potential window between −0.2 and 1.0 V with the scan rates ranging from 5 to 100 mV. GCD tests were carried out in the voltage range between −1.0 and 0 V. The gravimetric capacitances (F g$^{-1}$) were calculated through equation (2.1).

$$C = \frac{I\mathrm{d}V}{m\mathrm{d}t},$$ (2.1)

where $I$ (A) represents the current, ($\mathrm{d}V/\mathrm{d}t$) represents the slope of the discharge curves, and $m$ is the carbon mass of the electrodes.

Energy density ($E$, W h kg$^{-1}$) and power density ($P$, W kg$^{-1}$) were calculated by equations (2.1)–(2.3), respectively.

$$E = \frac{C}{8} \times (\Delta V - iR)^2$$ (2.2)

and

$$P = \frac{E}{\Delta t},$$ (2.3)

where $\Delta V$ (V) is the potential difference within the time $\Delta t$ (s), and $iR$ (V) is the voltage drop due to the inner resistance.

EIS tests were recorded at open-circuit voltage in the frequency range of 0.01–100 000 Hz with 10 mV alternating current amplitude.

## 3. Results and discussion

As for thermogravimetric (TG) and differential thermal analysis (DTA) curves of crude polysaccharide (figure 1), the sharp weight loss centred at 200°C was associated with the decomposition of the

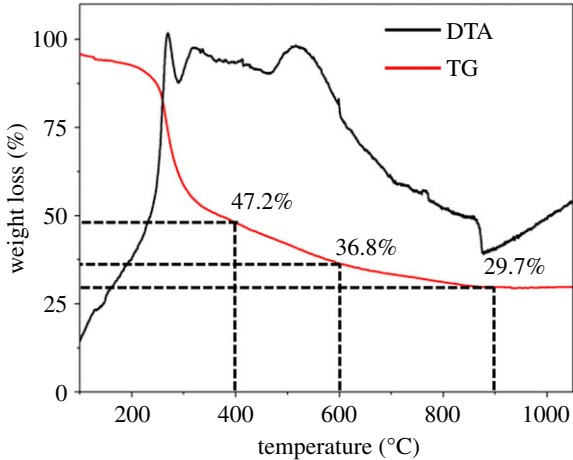

**Figure 1.** DTA and TG curves of the crude polysaccharide.

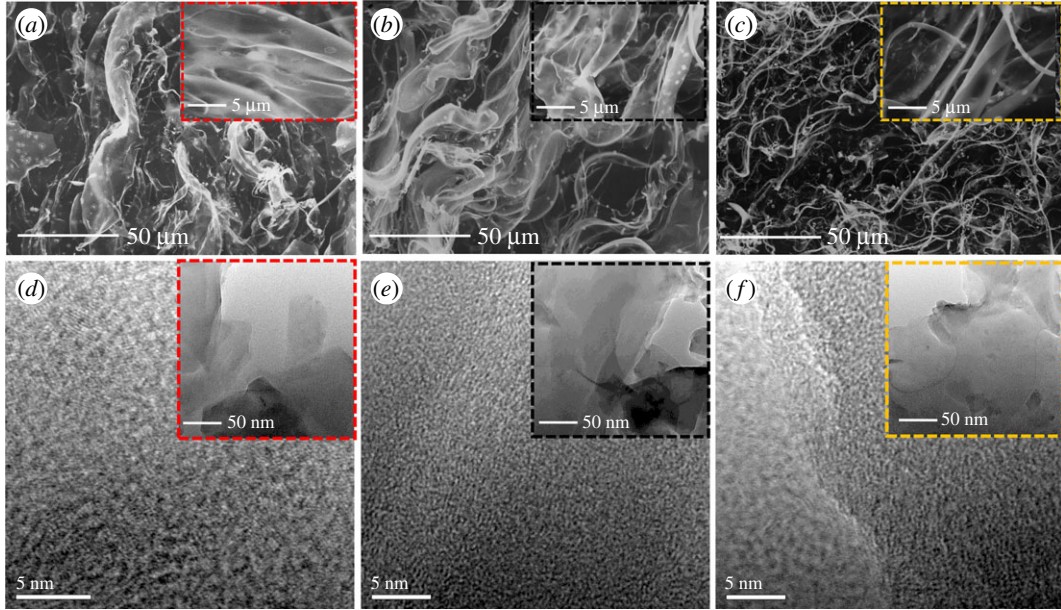

**Figure 2.** SEM images (a–c) of CPCs, TEM and HRTEM images (d–f) of CPCs. (a,d) CPC400, (b,e) CPC600, (c,f) CPC900.

organic matter in the polysaccharide [21]. After 400°C, the decomposition slowed down and stabilized after 900°C. Therefore, we determined the carbonization temperature to be 400, 600 and 900°C to analyse the effects of different temperatures on the microstructure and physical properties of CPCs. According to the TG curve, the weight loss of the crude polysaccharide was 70.3% at 900°C, which was much lower than the average level of biomass carbon studied by the predecessors, the lower weight loss could greatly reduce the production cost in future practical application. Elemental analysis and functional groups of the raw material and as-prepared material are shown in the electronic supplementary material, figure S1.

The morphology of the CPCs was studied by SEM and TEM. As shown in figure 2a–c, the CPCs show a natural sheet structure before 600°C.

After carbonization, a novel finding was that the CPC400–600 shows a three-dimensional smoke-like structure (figure 2a,b), TEM images (figure 2d–f) revealed that all the CPCs possess abundant micropores consisting of carbon materials and numerous disordered domains and low graphitization degrees [22]. However, the morphology was significantly destroyed when the temperature was increased to 900°C (figure 1c). The SEM images showed that as the temperature increases, the original structure of the material was greatly affected [23]. Three-dimensional smoke-like structure transformed into a fibrous structure from 600 to 900°C. The subsequent nitrogen adsorption isotherms further confirmed the effect of temperature on the original structure of carbon materials. The CPC400 and CPC600 samples exhibited large BET surface areas of 1920 and 2160 $m^2 g^{-1}$, with total pore volumes of

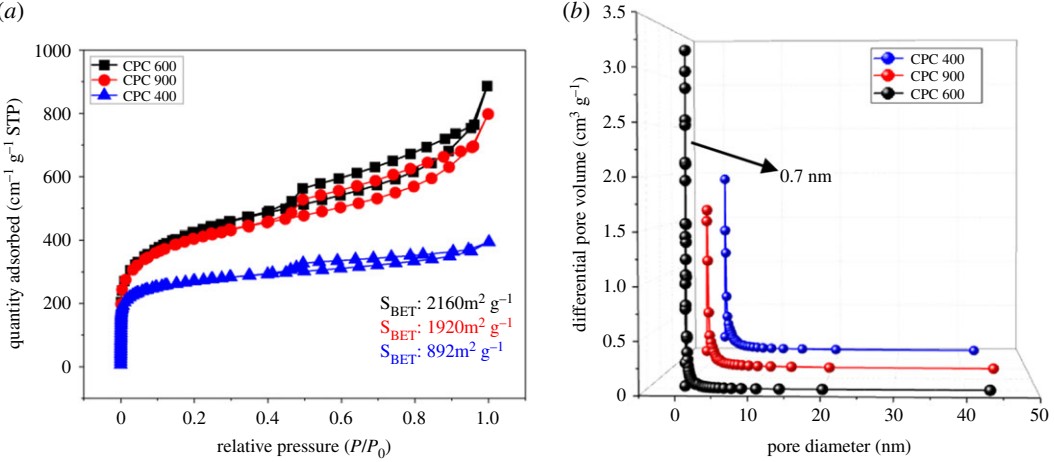

**Figure 3.** Nitrogen adsorption/desorption iostherms (*a*) and pore size distribution of CPCs (*b*).

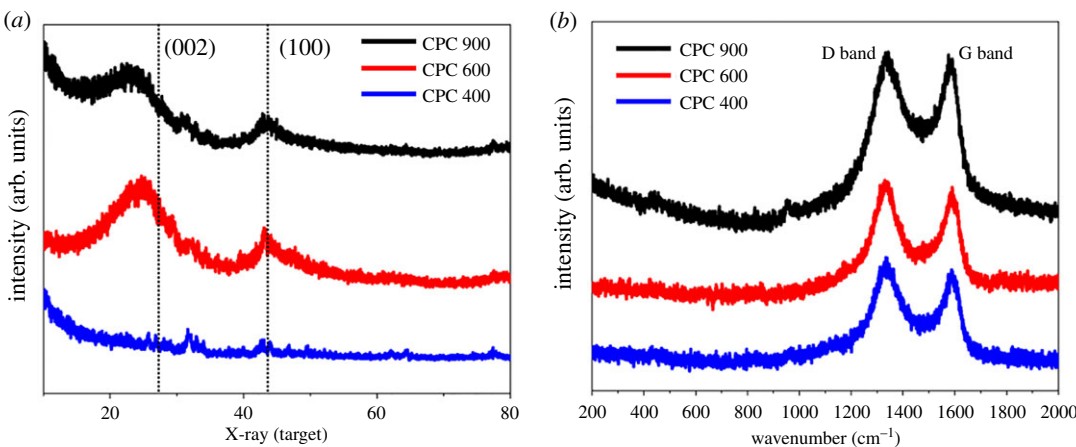

**Figure 4.** XRD patterns (*a*) and Raman spectrum (*b*) of CPCs.

**Table 1.** Texture properties of CPCs.

| sample | $S_{BET}^a$ ($m^2 g^{-1}$) | $S_{micro}^b$ ($m^2 g^{-1}$) | $S_{meso}^c$ ($m^2 g^{-1}$) | $V_t^d$ ($cm^3 g^{-1}$) |
|---|---|---|---|---|
| CPC400 | 892 | 503 | 103 | 0.42 |
| CPC600 | 2160 | 1203 | 230 | 1.35 |
| CPC900 | 1920 | 1012 | 349 | 1.21 |

[a]BET specific surface area.
[b]micropore specific surface area.
[c]mesopore specific surface area.
[d]total pore volume.

1.125 and 1.375 $cm^3 g^{-1}$, while the micropore volumes are 0.927 and 1.162 $cm^3 g^{-1}$ (calculated from the non-local density functional theory (NLDFT) method). As a comparison, the CPC900 sample shows a lower specific surface and a smaller total pore volume and micropore volume, as shown in figure 3. Additional adsorption parameters are summarized in table 1. The steep growth at low pressure indicates the presence of micropores, while the narrow hysteresis loop in the $P/P_0$ range of 0.5–1.0 indicates a certain amount of mesoporosity [24] (figure 3*a*). It should be noted that the curve for CPCs shows a peak value of approximately 0.70 nm (calculated from the Barrett–Joyner–Halenda (BJH) method) without any activator added [25], which was very consistent with the size of optimized ion-accessible sub-nanopores reported in other studies [26] (figure 3*b*); this result may be due to the activation of the K+ originally contained in the carbon precursor, which is one of our key research directions in the future. The large accessible surface area of CPC600, having pores that are mostly

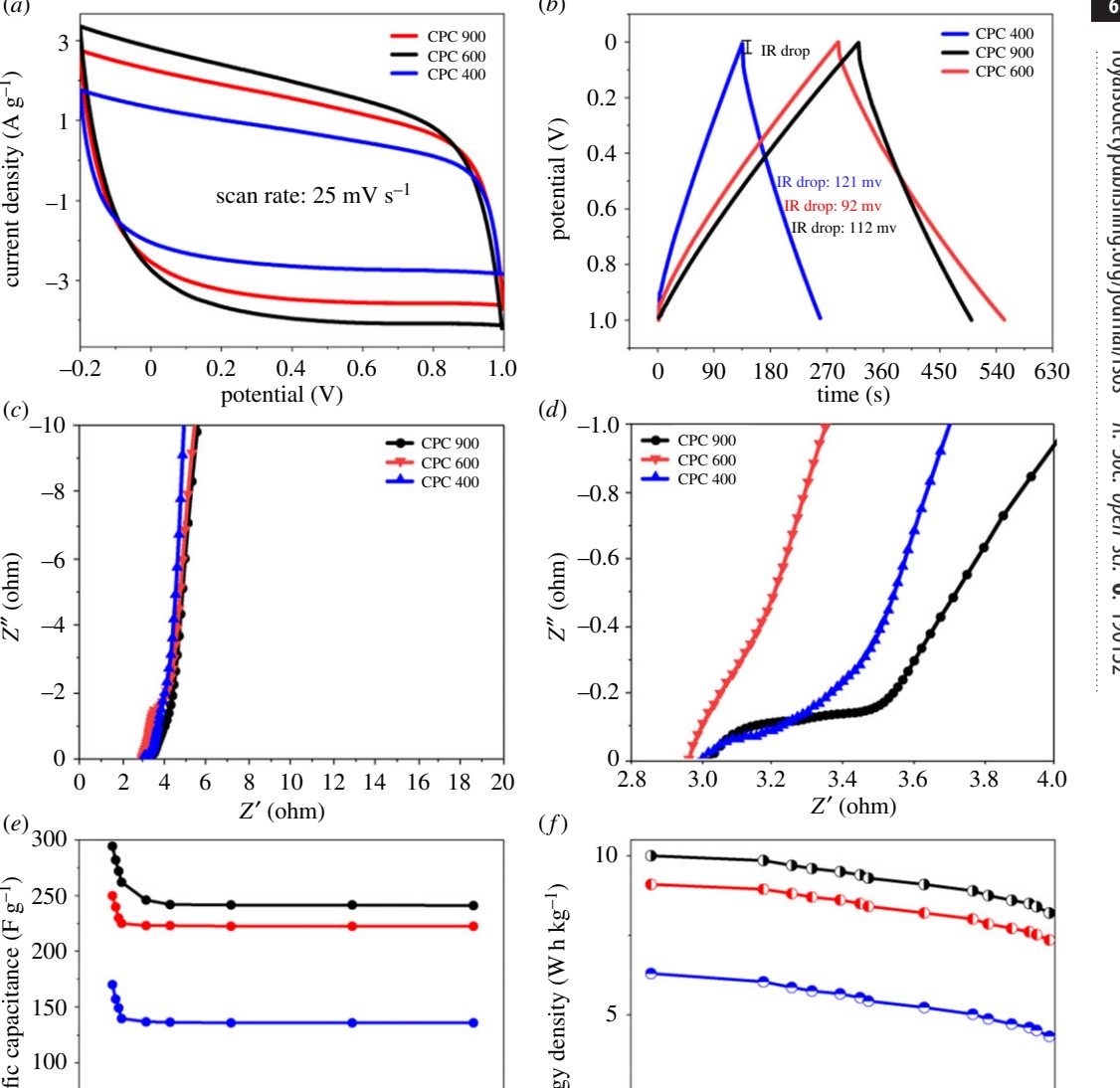

**Figure 5.** Electrochemical performance of CPCs. CV curves (*a*), GCD curves (*b*), Nyquist plots (*c,d*), specific capacitance under the current densities (*e*) and Ragone plots (*f*).

micropores, was beneficial for electrolyte penetration and ion adsorption, which might enhance the energy storage capability.

From the X-ray patterns, as shown in figure 4*a*, CPC400 and CPC600 demonstrate two diffraction peaks at $2\theta$ values of 27.2° and 44.3°, which are assigned to typical (002) and (100) reflections of graphitic carbon, respectively (JCPDF: 41–1487), while for the CPC900 sample, only a weak peak due to the (100) plane can be found, which demonstrates that the excessive temperature destroyed the graphitized structure of the material.

For Raman spectroscopy, as shown in figure 4*b*, the studied carbons all possessed numerous disordered domains and low graphitization degrees; typically, the D band at approximately 1350 cm$^{-1}$ relates to the disordered sp$^2$-hybridized carbon atoms of graphite or defect sites and the G band at approximately 1580 cm$^{-1}$ corresponds to the phonon mode for the in-plane vibration of sp$^2$-bonded carbon atoms, which is a typical symbol of graphitic carbon. The ID/IG band intensity ratio slightly decreased from 1.132 (CPC900) to 1.024 (CPC400), further illustrating the reduction in the degree of graphitization.

The electrochemical behaviour of the carbon materials was investigated by CV and GCD in a three-electrode system using 6 M KOH as the electrolyte [27]. It was clear that the voltage window of the CV

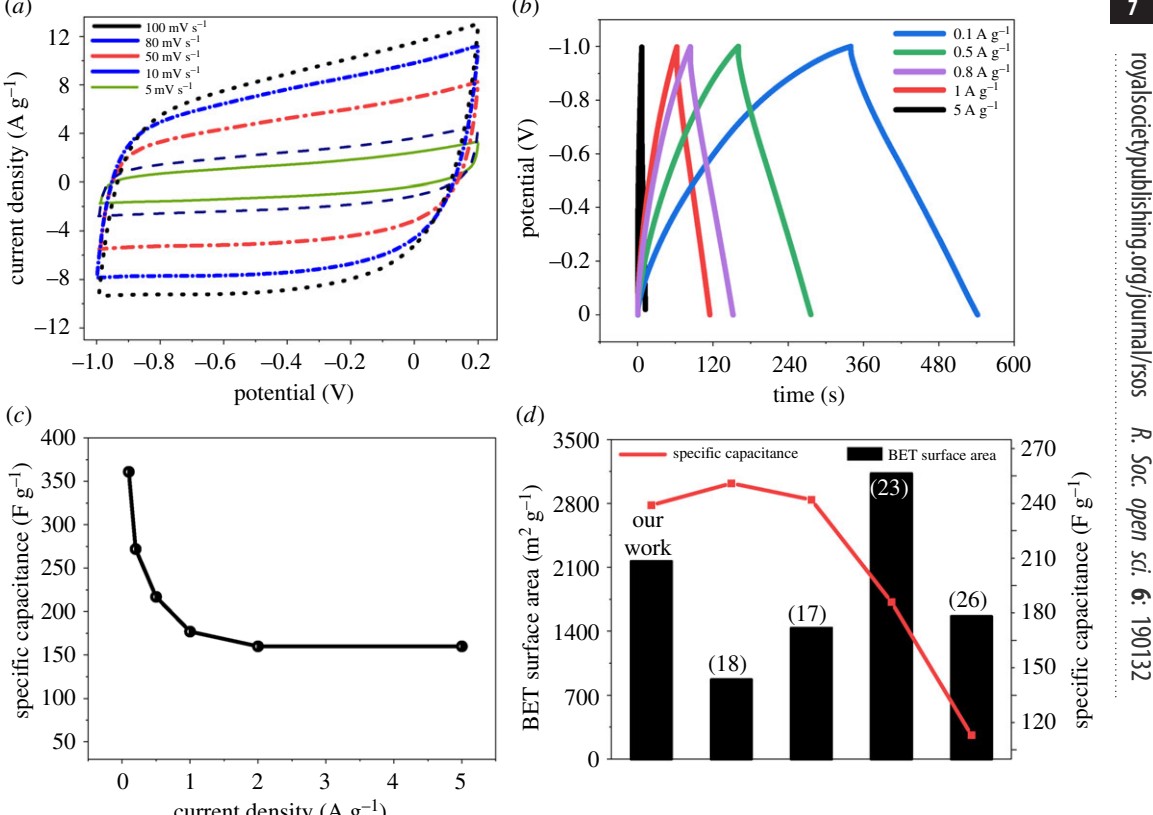

**Figure 6.** Electrochemical performance of CPC600 and the comparison with the previous work.

curve for the CPC400, CPC600 and CPC900 was still maintained even at 1.0 V, demonstrating that the supercapacitor can be reversibly cycled within the voltage window of −0.2 to 1.0 V (figure 5*a*). The CV curves of all the CPCs still retained a rectangular shape at the high scan rate (25 mV s$^{-1}$) indicating an excellent rate capability of the carbon materials. However, the slightly spindle-like shape of the CV curves for all the carbon samples indicated the presence of kinetic limitations of electrolyte ions entering smaller pores.

As for GCD tests in the two-electrode system, at 0.1 A g$^{-1}$ current density, GCD curves of the CPCs showed a perfect triangle with little obvious ohmic drops (figure 5*b*). This indicated rapid ion transport throughout the pores of the materials. The specific capacitances calculated from the GCD curves were between 292 and 174 F g$^{-1}$ at the low current density of 0.1 A g$^{-1}$ (figure 5*e*). The CPC900 sample exhibited a smaller specific capacitance, which demonstrates the effect of structural damage on the electrochemical performance of the carbon material. The self-discharge measurements of supercapacitors are shown in the electronic supplementary material, figure S3.

EIS was used to investigate the electron/ion transport process for the CPCs electrodes, as shown in figure 5*c,d*; the EIS plots show neat vertical straight lines in the low-frequency region, indicating the Warburg element ($W$) and low ion diffusion resistance. In the higher-frequency region, the small semicircle is indicative of charge transfer resistance and the low combined series resistance involving the intrinsic resistance of the electrode materials, ionic resistance of the electrolyte and contact resistance between the current collector and the electrode. The energy and power densities of the supercapacitors made from the CPCs were evaluated through the GCD curves in a two-electrolyte cell using 6 M KOH as an electrolyte. As shown in figure 5*f*, the CPCs' materials exhibited maximal densities of 6.8–10.2 W h kg$^{-1}$ under power outputs of 253.4 W kg$^{-1}$ to 24.3 kW kg$^{-1}$, which are clearly close to the values obtained from other carbon materials derived from biomass, while the carbon source of CPCs' materials came from cheaper wastes [28].

For real devices, the cyclic stability of the cell was not affected when the sweep rates were increasing. The results for CV, GCD for CPC600 are shown in figure 6. The CV profiles for the CPC600 electrode (figure 6) were measured under varying scan rates from 5 to 100 mV s$^{-1}$ between −0.2 and 1.0 V, which showed a symmetric rectangular shape with no broadened humps, indicating dominant

behaviour of the electrochemical double-layer capacitance (EDLC) with no pseudocapacitance from the oxygen-containing functional groups [29]. For GCD, the curves for the CPC600 electrode (figure 6) showed a triangular symmetry and small voltage drop due to the dominant EDLC behaviour; the capacitance for CPC600 was mainly attributed to its large surface area and pore volume [30]. As shown in figure 6, when the current density increased to $5 \, A \, g^{-1}$, the material retained the specific capacitance at $152 \, F \, g^{-1}$ (44.3%). It should be noted that the high specific capacitance of $152 \, F \, g^{-1}$ is remarkable for carbonaceous electrode materials without heteroatom (S, B or N) doping [3]. The cycle stability is shown in the electronic supplementary material, figure S2.

# 4. Conclusion

In summary, carbon electrode materials with a novel smoke-like structure were facilely synthesized from a bio-waste fungal substrate crude polysaccharide for use in high-performance supercapacitors. The as-prepared products show a large specific surface area ($2160 \, m^2 \, g^{-1}$) and excellent porosity, with a representative pore size of 0.7 nm; the material also shows a certain degree of graphitization and a flake-like structure. The specific capacitance for CPC600 reaches $361.0 \, F \, g^{-1}$ at $0.1 \, A \, g^{-1}$ and even retains a value of $152 \, F \, g^{-1}$ at $5 \, A \, g^{-1}$. Compared with the previous work (figure 6), CPCs' materials showed excellent specific surface area as well as ideal specific capacitance value (figure 6*d*). We hope that this method will lead to new ideas for the synthesis of future high-performance energy storage applications.

Data accessibility. This article does not contain any additional data.

Authors' contributions. M.C., Z.C. and M.L. designed the study, M.C., Z.H., J.C. and R.L. helped to finish the experiments, M.C. interpreted the results and wrote the manuscript. All authors gave final approval for publication.

Competing interests. There are no conflicts to declare.

Funding. This work was funded by the Jilin Province Science and Technology Development Projects (20190303120SF), Jilin Province Key Technology R&D Project (20180101212NY) and Changchun Science and Technology Project (18DY023).

Acknowledgements. We are grateful for instructional support by specialist Chao Zhang in Jilin Agricultural University; with her help we completed the antibacterial activities analysis. We also thank colleagues in our laboratory, Shang Wang, Jinshan Cao and Ye Zhou for help with the experiment, so that we could finish this work satisfactorily.

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
