## [Reviewer comments · Royal Society Open Science]

Review History

RSOS-190132.R0 (Original submission)

Review form: Reviewer 1

Is the manuscript scientifically sound in its present form?

Yes

Are the interpretations and conclusions justified by the results?

Yes

Is the language acceptable?

Yes

Is it clear how to access all supporting data?

Not Applicable

Do you have any ethical concerns with this paper?

No

Have you any concerns about statistical analyses in this paper?

I do not feel qualified to assess the statistics

Recommendation?

Major revision is needed (please make suggestions in comments)

Comments to the Author(s)

The authors synthesised the smoke-like carbon from a bio-waste fungal substrate crude polysaccharide and used as the electrode material of supercapacitors. The new finding of this work is in the synthesis part of this material. I recommend Accept after revision;

Comments

1. It is well known that biomass consists of impurities. The elemental analysis of raw materials and as-prepared materials e.g., WDXRF is needed.
2. For the electrochemical result, Coulombic and energy efficiencies are needed.
3. Self-discharge testing is also needed.
4. Any effects or charge contribution from the surface impurity or functional group?
5. The authors should cite other previous work relating to bio-activated carbon-based supercapacitors e.g., *Electrochimica Acta* 2019, 305, 443-451; *Electrochimica Acta* 2018, 286, 55-64.

Review form: Reviewer 2 (Mustafa S. Ata)**Is the manuscript scientifically sound in its present form?**

Yes

Are the interpretations and conclusions justified by the results?

Yes

Is the language acceptable?

Yes

Is it clear how to access all supporting data?

Yes

Do you have any ethical concerns with this paper?

No

Have you any concerns about statistical analyses in this paper?

No

Recommendation?

Accept with minor revision (please list in comments)

Comments to the Author(s)

Authors derived the smoke-like carbon for supercapacitors applications. The electrodes provide the specific capacitance as high as 152 F/g. Also the symmetric device showed good performance.

The manuscript is basically interesting. The problems should be clear before published.

1. BET calculation requires more details. Such as which device was used?
2. Miss-spell pg5 line 47 "ant" should be "and", also same page line 26 "respecticely"
3. Power density supposed to be calculated by discharge time, not time difference.
4. Please add "and" between 0.927 - 1.162 cm³/g; pg6 line 27.
5. Please add a,b,c,d to Figure 6.
6. I would like to see 1000 charge-discharge cycle performance for single electrode and symmetric device.
7. SEM pictures after the 1000 cycles would be beneficial.
8. Authors did not mention about the electrode preparation methods. What subtract material used, what was the mass loading of the electrodes?
9. Please try also asymmetric device, it would be interesting to see one side active carbon, carbon black or composites and the other side smoke-like carbon. Also provide 1000 cycles for the device as well.

Decision letter (RSOS-190132.R0)

10-May-2019

Dear Mr Chu:

Title: Novel Biomass-derived Smoke-like Carbon as a Supercapacitor Electrode Material
Manuscript ID: RSOS-190132

The editor assigned to your manuscript has now received comments from reviewers. We would like you to revise your paper in accordance with the referee and Subject Editor suggestions which can be found below (not including confidential reports to the Editor). Please note this decision does not guarantee eventual acceptance.

Please submit your revised paper before 02-Jun-2019. Please note that the revision deadline will expire at 00.00am on this date. If we do not hear from you within this time then it will be assumed that the paper has been withdrawn. In exceptional circumstances, extensions may be possible if agreed with the Editorial Office in advance. We do not allow multiple rounds of revision so we urge you to make every effort to fully address all of the comments at this stage. If deemed necessary by the Editors, your manuscript will be sent back to one or more of the original reviewers for assessment. If the original reviewers are not available we may invite new reviewers.

When submitting your revised manuscript, you must respond to the comments made by the referees and upload a file "Response to Referees" in "Section 6 - File Upload". Please use this to document how you have responded to the comments, and the adjustments you have made. In

order to expedite the processing of the revised manuscript, please be as specific as possible in your response.

On behalf of the Subject Editor Professor Anthony Stace and the Associate Editor Professor Tobias Hertel.

RSC Associate Editor:
Comments to the Author:
(There are no comments.)

RSC Subject Editor:
Comments to the Author:
(There are no comments.)

Reviewers' Comments to Author:
Reviewer: 1

Comments to the Author(s)

The authors synthesised the smoke-like carbon from a bio-waste fungal substrate crude polysaccharide and used as the electrode material of supercapacitors. The new finding of this work is in the synthesis part of this material. I recommend Accept after revision;

Comments

1. It is well known that biomass consists of impurities. The elemental analysis of raw materials and as-prepared materials e.g., WDXRF is needed.
2. For the electrochemical result, Coulombic and energy efficiencies are needed.
3. Self-discharge testing is also needed.
4. Any effects or charge contribution from the surface impurity or functional group?
5. The authors should cite other previous work relating to bio-activated carbon-based supercapacitors e.g., *Electrochimica Acta* 2019, 305, 443-451; *Electrochimica Acta* 2018, 286, 55-64.

Reviewer: 2

Comments to the Author(s)

Authors derived the smoke-like carbon for supercapacitors applications. The electrodes provide the specific capacitance as high as 152 F/g. Also the symmetric device showed good performance. The manuscript is basically interesting. The problems should be clear before published.

1. BET calculation requires more details. Such as which device was used?
2. Miss-spell pg5 line 47 "ant" should be "and", also same page line 26 "respecticely"
3. Power density supposed to be calculated by discharge time, not time difference.
4. Please add "and" between 0.927 - 1.162 cm³/g; pg6 line 27.
5. Please add a,b,c,d to Figure 6.
6. I would like to see 1000 charge-discharge cycle performance for single electrode and symmetric device.
7. SEM pictures after the 1000 cycles would be beneficial.
8. Authors did not mention about the electrode preparation methods. What subtract material used, what was the mass loading of the electrodes?
9. Please try also asymmetric device, it would be interesting to see one side active carbon, carbon black or composites and the other side smoke-like carbon. Also provide 1000 cycles for the device as well.

Author's Response to Decision Letter for (RSOS-190132.R0)

See Appendix A.

Decision letter (RSOS-190132.R1)

24-Jun-2019

Dear Mr Chu:

Title: Novel Biomass-derived Smoke-like Carbon as a Supercapacitor Electrode Material
Manuscript ID: RSOS-190132.R1

It is a pleasure to accept your manuscript in its current form for publication in Royal Society Open Science. The chemistry content of Royal Society Open Science is published in collaboration with the Royal Society of Chemistry.

On behalf of the Subject Editor Professor Anthony Stace and the Associate Editor Professor Tobias Hertel.

RSC Associate Editor
Comments to the Author:
(There are no comments.)

Reviewer(s)' Comments to Author:

Appendix A

Dear Editor and Reviewers:

Thank you for your letter and for the reviewers' comments concerning our manuscript entitled *Novel Biomass-derived Smoke-like Carbon as a Supercapacitor Electrode Material*.

Those comments are all valuable and very helpful for revising and improving our paper, as well as the important guiding significance to our researches. We have studied comments carefully and have made correction which we hope meet with approval. The main corrections in the paper and the responds to the reviewer's comments are as following:

Coments:

Reviewer1:

- 1 , It is well known that biomass consists of impurities. The elemental analysis of raw materials and as-prepared materials.
- 2 , For the electrochemical result, Coulombic and energy efficiencies are needed.
- 3 , Self-discharge testing is also needed
- 4 , Any effects or charge contribution from the surface impurity or functional group?
- 5 , The authors should cite other previous work relating to bio-activated carbon-based supercapacitors e.g., *Electrochimica Acta* 2019, 305, 443-451; *Electrochimica Acta* 2018, 286, 55-64.

Review2:

1. BET calculation requires more details. Such as which device was used?
2. Miss-spell pg5 line 47 "ant" should be "and", also same page line 26 "respecticely"
3. Power density supposed to be calculated by discharge time, not time difference.
4. Please add "and" between 0.927 - 1.162 cm³/g; pg6 line 27.
Please add a,b,c,d to Figure 6.
5. I would like to see 1000 charge-discharge cycle performance for single electrode and symmetric device.
6. SEM pictures after the 1000 cycles would be beneficial.
7. Authors did not mention about the electrode preparation methods. What subtract material used, what was the mass loading of the electrodes?
8. Please try also asymmetric device, it would be interesting to see one side active carbon, carbon black or composites and the other side smoke-like carbon. Also provide 1000 cycles for the device as well.

Response to reviewer1:

1. It is well known that biomass consists of impurities. The elemental analysis of raw materials and as-prepared materials.

We are very sorry for our unclear report in the basic characterization of raw material and as-prepared materials. For the electrode material of supercapacitor, the influence of its elemental composition on its EDLC behavior is enormous. The elemental analysis is needed. As shown in Fig.1a, the crude polysaccharide has a natural three-dimensional

Fig. 1 (a),(b): The photograph of the raw material (polysaccharide). (c): FT-IR analysis of the polysaccharide and CPC600. (d) The edx analysis of polysaccharide and CPC600. (e), (f), (g), (h): The mapping analysis of the CPC600

self-supporting mechanism. Before carbonization, in addition to the common elements in organic substances such as C, O, the polysaccharide also contains some K

elements, and there are no other impurities (such as heavy metals such as Fe) that affect its electrochemical properties, and these naturally occurring activators It may also be the reason why the prepared material desired specific surface area and porosity. After carbonization, as shown in fig.1b, the

2. For the electrochemical result, Coulombic and energy efficiencies are needed.

Fig 2. The specific capacitance and coulombic efficiency of CPC600

As shown in Fig S2(b), after 5000 cycles, the CPC600 has a certain agglomeration phenomenon, and its flaky structure was not obviously destroyed after thousands of cycles. Therefore, the specific capacitance of CPC600 still retains 92.1% after 5000 cycles, as shown in fig s2(a). As for the coulombic efficiency, after 5000 cycles, the coulomb efficiency is generally stable at 90%, which indicates that CPC has good cycle stability.

3. Self-discharge testing is also needed.

Fig 3. The self-discharge measurements of supercapacitors

As shown in fig.3, the self-discharge measurements of supercapacitors were carried out according to the IEC 62391-1 standard. The supercapacitors were completely discharged prior to the self- discharge study. Then, the supercapacitors were charged up to a voltage of 2.3 V with a constant current so that 95% of their rated

$$EPR = \frac{(t_2 - t_1)}{\ln(V_2 - V_1) \times C}$$

voltage is attained within 30 minutes. It was held at rated voltage for 8 hours. After the 8 hours holding period, it was then disconnected from the voltage source. The open circuit voltage of supercapacitor was recorded using the Supercapacitor Testing System for 16 hours. The Equivalent Parallel Resistance (EPR) of

supercapacitors was calculated from the open circuit voltage of the supercapacitors using equation (1):

(1)

Where V_1 and V_2 are the voltages at time t_1 and t_2 respectively, C is the

$$I_L = C \times \frac{dV}{dt}$$

capacitance of supercapacitor in Farad. The leakage current of the supercapacitors was also calculated using (2)

(2)

Where I_L is the leakage current, dV/dt is the slope of the curve and C is the capacitance

in Farad. Fig S2a shows the time dependent decrease in open circuit potential of graphene supercapacitors. It is obvious from fig S2 that the open circuit voltages of the supercapacitors are decreasing with respect to time. The instantaneous initial drop is due to relatively higher Equivalent Series Resistance (ESR) of graphene supercapacitors. After the instantaneous drop due to ESR, we can observe two distinct portions in the self-discharge curve of which the first one is a fast discharge portion with a shorter time duration followed by a slow discharge portion. The mechanism of self-discharge can be understood by analysing the selfdischarge curve in detail. Fig S2 (c) shows the relation between the log (self-discharge voltage) as a function of time. Generally, discharge through an ohmic leakage leads to a declining linear relation between $\log V$ and t . From Fig S2(c), it is evident that, even after

neglecting the initial drop due to ESR, the curve does not follow a perfect linear path. Hence the self-discharge cannot be only due to the ohmic leakage pathways between the two electrodes. There can be some additional mechanism which also contributes to the self-discharge. In order to understand whether there is any diffusion-controlled mechanism, we have also plotted V vs $t^{1/2}$. Fig S3(d) shows the relation between self-discharge voltage decline V and $t^{1/2}$. In general, due to diffusion controlled faradaic leakage current in carbon based supercapacitors, the open circuit voltage shows a linear declining relation with $t^{1/2}$. As obvious from Fig S3(d), the V and $t^{1/2}$ have a better linear relation particularly in the slow discharge region. This clearly indicates that the diffusion-controlled mechanism will be the predominant mechanism in self discharge of these supercapacitors in addition to the ohmic leakage[1]. From fig S3(c), it is clear that V vs $\log t$ curves shows no linearity. Thus it can be confirmed that there is no contribution from overcharging in the self-discharge of graphene supercapacitors. Hence it is clear that the self-discharge in these graphene supercapacitors is controlled by the combined contribution from potential controlled model due to ohmic leakage and diffusion-controlled model due to charge re-distribution phenomenon.

4. Any effects or charge contribution from the surface impurity or functional group?

As for FT-IR for the some-like carbon, the broad peak at 3500 cm^{-1} is the intermolecular and intramolecular -OH group stretching vibration peak of the fungus polysaccharide, and the double peak at 2900 cm^{-1} is the CH₂ group in the fiber, and the peak at 1650 cm^{-1} indicates C=O. The symmetric stretching vibration, the peak at

1100 cm^{-1} is the stretching vibration of the C-O-C group. The peak at 1000 cm^{-1} indicates the presence of a pyranose ring in the polysaccharide of the fungus. After carbonization and pickling, potassium ions almost completely evaporate, so the existence of potassium is one of the key points of our further study [2].

5. The authors should cite other previous work relating to bio-activated carbon-based supercapacitors e.g., *Electrochimica Acta* 2019, 305, 443-451; *Electrochimica Acta* 2018, 286, 55-64.

We are very glade to quote these two works in this manuscript. Sethuraman Sathyamoorthia studied a simple and practical hybrid dual ionic liquid (IL)/1.0 M LiTFSI(aq.) is proposed using ILs in the nanopores of activated carbon for high cell potential supercapacitor and Nutthaphon Phattharasupakun have found that a fast Na ion diffusion of 10^{-8} - 10^{-11} $\text{cm}^2 \text{ s}^{-1}$ and a fast-standard heterogeneous rate constant of electron transfer of ca. 10^{-5} cm s^{-1} are two reasons leading to high-performance NIC. Give us a lot of help for our work on carbon-based supercapacitor testing, especially in self-discharge testing and cycle stability testing.

Response to reviewer2:

1. BET calculation requires more details. Such as which device was used?

Fig 1. BET analysis (a) and pore size distributions (b) of CPCs

To investigate the porous structure and surface area of CPC, N₂ adsorption/desorption was carried out as shown in FIG.S4. The CPC exhibits type I isotherm with a small H4 hysteresis loop indicating the existence of microporous/mesoporous with small amount of microporous. Materials. Note, the CPCs exhibits very high N₂ uptake in the low-pressure region ($p/p_0 < 0.01$), suggesting the very large amounts of microporosity. The calculated BET specific surface area, total pore volume, and mean pore diameter of CPC are 2377 m² g⁻¹, 1.50 cm³ g⁻¹, and 2.53 nm, respectively. For the pore size distribution in the micropore region (calculated from NLDFT method) as shown in Fig S4, the CPC displays a single pore size of 1.93 nm, whilst in the mesopore region (calculated from BJH method), the average pore size is 0.7 nm as shown in the inset image.

2. Miss-spell pg5 line 47 "ant" should be "and", also same page line 26 "respectively".

We are very sorry for the confusion for editors and reviewers caused by our negligence in the writing process. We have already checked the manuscript and corrected the above mistakes.

3. Power density supposed to be calculated by discharge time, not time difference.

Thanks to the reviewer for pointing out, we are sorry for the unclear report in the manuscript. The Δt means the discharge time instead of difference, we have changed Δt (s) to t (s) in our manuscript to make it more clear.

4. Please add "and" between 0.927 - 1.162 cm³/g; pg6 line 27. Please add a,b,c,d to Figure 6.

Fig.2. Electrochemical performance of CPC600 and the comparison with the previous

work.

We are very sorry for the confusion for editors and reviewers caused by our negligence in the writing process. We have already checked the manuscript and corrected the above mistakes.

5. I would like to see 1000 charge-discharge cycle performance for single electrode and symmetric device.

Fig 3. The specific capacitance and coulombic efficiency of CPC600

As shown in Fig 3b, after 5000 cycles, the CPC600 has a certain agglomeration phenomenon, and its flaky structure was not obviously destroyed after thousands of cycles. Therefore, the specific capacitance of CPC600 still retains 92.1% after 5000 cycles, as shown in fig 3(a). As for the coulombic efficiency, after 5000 cycles, the coulomb efficiency is generally stable at 90%, which indicates that CPCs has good cycle stability^[4].

6. SEM pictures after the 1000 cycles would be beneficial.

As shown in Fig 3, after 5000 cycles, the CPC600 has a certain agglomeration phenomenon, and its flaky structure was not obviously destroyed after thousands of cycles. Therefore, the specific capacitance of CPC600 still retains 92.1% after 5000 cycles, as shown in fig s2(a). As for the coulombic efficiency, after 5000 cycles, the coulomb efficiency is generally stable at 90%, which indicates that CPC has good cycle

stability.

7. Authors did not mention about the electrode preparation methods. What substrate material used, what was the mass loading of the electrodes?

We are very sorry for our unclear report on the preparation methods of the electrodes and the solid-state symmetric supercapacitors, now add as follows:

1, Fabrication of electrodes and solid-state symmetric supercapacitors

Nickel foam was first cut into rectangle sheets (20 mm * 10 mm) and treated with acetone, diluted HCl and deionized water each for 10 min ultra-sonication. A mixture containing 80 wt% active material, 10 wt% conductive carbon black and 10 wt% polyvinylidene fluoride (PVDF) was well grinded with appropriate amount of N-methyl-2-pyrrolidone (NMP) for 1 h to obtain a black paste. The paste was then casted on half of the pre-treated nickel foam and dried in a vacuum oven at 120 °C for 8 h. The electrode was finally obtained by pressing at 10 MPa for 1 min. The loading mass of active materials on each working electrode was about 3.0 mg. The solid-state symmetric supercapacitor was assembled by two identical electrodes, wherein KOH/polyvinyl alcohol (PVA) gel was used as the electrolyte. For preparing the KOH/PVA gel, 2.0 g PVA was first mixed with 20 mL deionized water and the mixture was heated to 85 °C under vigorous stirring until it became clear. Then, 10 mL KOH solution (6.0 M) was slowly added into the above mixture. The solution was kept stirring for 0.5 h at 85 °C to form a clear gel electrolyte. Two electrodes were immersed in the as-prepared electrolyte for 5 min before assembly. Then, the electrodes were picked out and transferred to a fume hood at room temperature for 1 h to vaporize the excess water. Finally the electrodes were pressed together under the pressure of 1 MPa for

10 min and sealed with plastic wrap to assemble the solid-state supercapacitor. The total mass of active materials for a symmetric supercapacitor was 6.0 mg^[3].

2, Fabrication of coin-type symmetric supercapacitors in ionic liquid electrolyte

The electrochemical performances of the PGBC-based symmetric supercapacitors in ionic liquid electrolyte were measured in a two-electrode cell configuration (CR2032-type coin cell). The electrodes were prepared by coating the aforementioned mixture containing active materials onto current collectors (nickel foam) with loading mass of about 8 mg/cm², then dried in vacuum at 120 °C for 8 h and pressed at 10 MPa. A neat ionic liquid of 1-Ethyl-3methylimidazolium bis(trifluoromethylsulfonyl)imide (EMIM TFSI) was used as the electrolyte, and a polypropylene membrane (MPF30AC, NKK, Japan) as the separator. The coin-type supercapacitors were finally assembled in an argon-filled glove box.

- 8. Please try also asymmetric device, it would be interesting to see one side active carbon, carbon black or composites and the other side smoke-like carbon. Also provide 1000 cycles for the device as well.**

Thanks to the reviewers for making such interesting suggestions and making our research more comprehensive. We have tried to assemble the activated carbon and acetylene respectively with CPC600s into devices and studied their electrochemical properties. We are sorry that due to the mistakes in the experimental operation, we can't finish the data before June 2. We will study the cyclic stability of asymmetric devices in future research and looking forward to further communication with you.

Notes and references

- [1] L. Shi, Li. Jin, Z. Meng, Y. C. Li and Y. Shen, *RSC Adv.* 2018, **8**, 39937-39947
- [2] Y. Gong, D. L. Li, C. Z. Luo, Q. Fu, C. X. Pan, *Green Chem.* 2017, **19**, 4132-4140
- [3] M. Zhou, Y. Lu, H. Chen, X. Ju, F. Xiang, *Journal of Ener stor*, 2018, **19**, 35-40
- [4] K. Gao, Q. Tang, Y. Guo, L. Wang, *Journal of Elec Materi*, 2018, **47**, 337-346

We sincerely appreciate your insightful and constructive comments and suggestions. We appreciate for Editor/Reviewers' warm work earnestly, and hope that the correction will meet with approval.

Once again, thank you very much for taking the time to review this paper.

Sincerely,

Mingxu Chu